# Core-Shell Structured PLGA Particles Having Highly Controllable Ketoprofen Drug Release

**DOI:** 10.3390/pharmaceutics15051355

**Published:** 2023-04-28

**Authors:** Norbert Varga, Rita Bélteki, Ádám Juhász, Edit Csapó

**Affiliations:** 1MTA-SZTE Lendület “Momentum” Noble Metal Nanostructures Research Group, University of Szeged, Rerrich B. Sqr. 1, H-6720 Szeged, Hungary; vargano@chem.u-szeged.hu (N.V.); juhaszad@chem.u-szeged.hu (Á.J.); 2Interdisciplinary Excellence Center, Department of Physical Chemistry and Materials Science, University of Szeged, Rerrich B. Sqr. 1, H-6720 Szeged, Hungary

**Keywords:** PLGA, ketoprofen, core-shell nanocarrier, controlled release

## Abstract

The non-steroid anti-inflammatory drug ketoprofen (KP) as a model molecule is encapsulated in different poly(lactide-co-glycolide) (PLGA) nanostructured particles, using Tween20 (TWEEN) and Pluronic F127 (PLUR) as stabilizers to demonstrate the design of a biocompatible colloidal carrier particles with highly controllable drug release feature. Based on TEM images the formation of well-defined core-shell structure is highly favorable using nanoprecipitation method. Stabile polymer-based colloids with ~200–210 nm hydrodynamic diameter can be formed by successful optimization of the KP concentration with the right choice of stabilizer. Encapsulation efficiency (EE%) of 14–18% can be achieved. We clearly confirmed that the molecular weight of the stabilizer thus its structure greatly controls the drug release from the PLGA carrier particles. It can be determined that ~20% and ~70% retention is available with the use of PLUR and TWEEN, respectively. This measurable difference can be explained by the fact that the non-ionic PLUR polymer provides a steric stabilization of the carrier particles in the form of a loose shell, while the adsorption of the non-ionic biocompatible TWEEN surfactant results in a more compact and well-ordered shell around the PLGA particles. In addition, the release property can be further tuned by decreasing the hydrophilicity of PLGA by changing the monomer ratio in the range of ~20–60% (PLUR) and 70–90% (TWEEN).

## 1. Introduction

Increasing of potential drug molecules’ efficiency and the promoting of their targeted delivery processes are a key research area in the biomedical and pharmaceutical developments [1,2]. Due to the structure-tunable physical/chemical properties, many types of nano- or micro-sized colloidal particles can be applied for the formulation of active ingredients [3,4,5]. The biodegradable and biocompatible hydrophobic polymers, like poly(lactic acid) (PLA) [6,7], polycaprolactone (PCL) [8], poly(trimethylene carbonate) (PTMC) [9] or polyurethanes (PUR) [10] are widely used. The hydrophilicity of these polymers can be greatly controlled by the copolymerization thus making it feasible to produce a carrier that can be tailored to the character of the active compound to be encapsulated [11]. Among these polymers, the poly(lactide-co-glycolide) (PLGA) copolyester is frequently used biodegradable polymer [12]. PLGA-based nanoparticles (NPs) can be prepared by nanoprecipitation [13,14], emulsion solvent evaporation [15,16,17], nano spray drying [18], dialysis [19] or supercritical fluid [20] methods, where the type of the organic solvent, stabilizer, or the monomer ratios can be significantly influenced the properties (e.g., size, structure, controlled drug release) of the formed particles [21,22]. The polyethylene glycol and polypropylene glycol triblock copolymer Pluronic F127 (PLUR) and the polysorbate -type non-ionic surfactant Tween 20 (TWEEN) are often applied non-toxic stabilizers (Oral (rat) LD50: ~5 g/kg (Pluronic F127) [23]; no toxicity (rats): 0.4 mg/kg (Tween20) [24]), which can be used in the field of the cosmetics, food, or pharmaceutical (as delivery of the proteins, peptides or different drugs etc.) applications. Ketoprofen (KP) (2-(3-benzolphenyl) propionic acid) is a nonsteroid anti-inflammatory drug (NSAID) from the class of the propionic acid with antipyretic and analgesic features, which can be used to treat musculoskeletal, joint disorders, migraine headache or infectious disease etc. [25]. Many types of KP-loaded PLGA nano-(NPs) and microparticles (MPs) can be found in the literature, but their uniform interpretation and comparison are difficult due to the various measurement conditions. Owen I. Corrigan and Xue Li encapsulated NSAID (KP, indomethacin) molecules in PLGA50 by emulsion solvent evaporation method, where ~4.2% drug loading (DL%) and ~40.2% encapsulation efficiency can be achieved with ~2 µm size span [26]. Changhui Yu et al. prepared micro-sized PLGA carriers containing KP and microRNA (miR-124). After the formation of the particles by W/O/W emulsion technique, they could increase the DL% to ~14.5%, but the particles were ~47.4 µm [27]. Johannes Kluge and co-workers characterized the time-dependence of the KP content for the PVA-stabilized PLGA micro- and nanocarriers with 100–200 nm in size, and they found that, ~5% drug loading can be achieved with the racemic KP crystals [20]. However, this value is increased to 10% without crystals. Important to note that the exact hydrodynamic diameters and the release were not presented in the publication. Elisabetta Gavini et al. used spray-dried technique to synthesize PLGA microsphere (10–77 µm), but no information available for the value of EE% and DL% for KP [28]. In our previous work, PLA- and various PLGA-based nanosized particles were prepared to determine the encapsulation possibilities of several drugs having different solubility (KP, (±)-α-tocopherol (TP) and D-α-tocopherol polyethylene glycol 1000 succinate). Moreover, for each system we have determined the dominant effect of the experimental conditions like the type of the organic solvent (acetone, 1,4-dioxane) and the stabilizer (hexadecyltrimethylammonium bromide, Pluronic F127, polyvinyl alcohol) on the size as well as the structure of the formed particles [29]. Well-defined core-shell structure can be formed in case of the more hydrophobic drugs (as TP), and the characteristic structure can be systematically changed by the concentration of the main components, but this was not studied previously for KP [30]. 

Based on these results, in this work we mainly focus on the preparation of core-shell type KP-loaded PLGA particles with nanosized structure using nanoprecipitation method, where the previously not investigated synthesis conditions like PLGA copolymer composition, KP concentration or the type of the stabilizers are investigated on the size, structure and EE% and DL% values. In addition, we studied the in vitro release profiles of the drug-contained different carriers at several pH and temperature, where the type of the applied stabilizers and the monomer ratios of the PLGA strongly influence the release properties. 

## 2. Materials and Methods

### 2.1. Materials

The poly(lactide-co-glycolide) copolymers with different lactide to glycolide ratio (PLGA50: 50/50, M_w_ = 30,000–50,000 Da; PLGA65: 65/35, M_w_ = 40,000–75,000 Da; PLGA75: 75/25, M_w_ = 66,000–107,000 Da) were purchased from Sigma-Aldrich (Budapest, Hungary). Sodium phosphate dibasic dodecahydrate (Na_2_HPO_4_ × 12H_2_O; ≥99%), sodium phosphate monobasic dihydrate (NaH_2_PO_4_ × 2H_2_O; ≥99%), sodium chloride (NaCl; ≥99%), acetone (C_3_H_6_O; ≥99%) and dimethyl-sulfoxide (DMSO; C_2_H_6_OS; ≥99%) were obtained from Molar Chemicals (Budapest, Hungary). The ketoprofen (KP, C_16_H_14_O_3_, ≥98%), Pluronic F127 (PLUR, (C_3_H_6_O·C_2_H_4_O)x, M_w_ = ~12,600 Da) and Tween20 (TWEEN, C_58_H_114_O_26_, M_w_ = 1228 Da) were obtained from Sigma-Aldrich. Highly purified water was obtained with a Millipore Direct-Q 3 UV purification apparatus (18.2 MΩ·cm at 25 °C). The components and solvents were analytical grade and further purifications were not used.

### 2.2. Methods

#### 2.2.1. Preparation of the Drug-Free and KP-Loaded PLGA NPs

For the preparation of the PLGA50 drug carriers, nanoprecipitation method was used under different initial parameters: c_PLUR/TWEEN_ = 0.0–1.0 mg/mL, c_PLGA_ = 2.5–10.0 mg/mL, c_NaCl_ = 0.0–0.16 M. During the formation of the drug-free particles, the PLGA50 was dissolved in organic solvent (acetone, or DMSO), and 0.5 mL stock solution was added dropwise to 5 mL stabilizer (PLUR or TWEEN)-contained aqueous phase (organic phase to aqueous phase ratio 1:10). The samples were stirred at 350 rpm at room temperature for a day and after that the prepared PLGA50 particles were stored at 10 °C until the further used. 

In case of the KP-loaded PLGA systems, firstly the copolymer derivatives (PLGA50/65/75) (c_PLGA_ = 10 mg/mL) were dissolved in acetone making individual samples which also contained different concentrations of dissolved KP (c_KP_ = 0–15 mg/mL). Secondly, 0.5 mL of this organic phase was dropped to an aqueous TWEEN or PLUR stabilizer-containing solution (V = 5 mL, c = 0.1 mg/mL) which results in the precipitation of PLGA particles. To eliminate the non-encapsulated KP molecules the dispersions were centrifuged (12,000 rpm, 15–20 min) and the supernatant was removed. Dispersions were freeze-dried (using Christ Alpha 1–2 LD plus equipment) to obtain solid samples for further structural characterizations. The solid samples were stored at −20 °C.

#### 2.2.2. Methods for Size, Size Distribution and Structural Characterizations

The transmission electron microscopy (TEM) images were recorded by Jeol JEM-1400plus equipment (JEOL Ltd., Tokyo, Japan) using 120 keV accelerating voltage. The particle size, size distribution and the ζ-potential values of the drug-free PLGA and KP-loaded PLGA particles were determined with a Horiba Sz-100 (HORIBA Jobin Yvon, Longjumeau, France) dynamic light scattering (DLS) equipment, which has a diode pumped frequency doubled (532 nm, 10 mW) laser. For evaluation of the DLS data both the refractive indexes and the viscosities of the solvent mixtures were determined. The parallel experiments were performed with 90° of detection angle at 25 ± 0.1 °C. To determine the encapsulation efficiency (EE%) and the drug loading (DL%) of the carriers, Shimadzu UV-1800 UV-Vis double beam spectrophotometer was used. The lyophilized drug-loaded carriers were dissolved in 1,4-dioxane and sonicated for 30 s. The UV-Vis spectra of the samples were registered in the range of 200–400 nm using 1 cm quartz cuvette at room temperature, and the concentration of the KP was calculated from the calibration curve (Appendix A). After the purification of the samples the amount of the non-encapsulated drug molecules in the supernatant was also determined by this way. The characteristic absorbance band was detected at 252 nm (in 1-4-dioxane). The EE% (Equation (1)) and the DL% (Equation (2)) were determined by the initial drug mass used in the formulation and the total mass of the NPs, respectively.
(1)$$\mathrm{EE}\%=\frac{\mathrm{encapsulation}\;\mathrm{mass}\;\mathrm{of}\;\mathrm{drug}}{\mathrm{total}\;\mathrm{mass}\;\mathrm{of}\;\mathrm{drug}}\;\times \;100$$
(2)$$\mathrm{DL}\%=\frac{\mathrm{encapsulation}\;\mathrm{mass}\;\mathrm{of}\;\mathrm{drug}}{\mathrm{total}\;\mathrm{mass}\;\mathrm{of}\;\mathrm{nanoparticles}}\;\times \;100$$

Differential scanning calorimetry (DSC) curves of the solid samples were recorded by Mettler Toledo DSC822e in the temperature range of 25–500 °C, where nitrogen with 50 mL/min flow rate was used. The heating rate was 5 °C/min in every cases. FT-IR spectra of the KP-loaded PLGA systems were obtained by a Jasco FT/IR-4700 with ATR PRO ONE Single-reflection accessory (ABL&E-JASCO, Budapest, Hungary) at room temperature. The measurements were carried out in 500–4000 cm^−1^ wavenumber range with 1 cm^−1^ resolution at room temperature. The spectra were determined by 128 interferograms.

#### 2.2.3. In Vitro Release Studies

The in vitro release curves were obtained by UV-Vis spectrophotometer, where the dissolution profiles were registered at different pH (pH = 6, 7.4 and 8; phosphate buffer with 0.9% NaCl) and temperatures (T = 25, 37 and 45 °C). The quantity of the active substance released was determined by calibration curves at λ = 260 nm (in PBS medium) (Appendix A). The pure drug and the drug-contained samples were placed separately in a cellulose (dialysis) membrane (M_w cut-off_ = 14,000 Da; Sigma-Aldrich) with 5 mL buffer solution, which were inserted into 35 mL dissolution medium, and the penetration of the KP molecules was followed for 420 min. The dissolution curves were fitted by several kinetic equations (first order, second order, Higuchi, Weibull and Korsmeyer-Peppas models) using nonlinear regression [31,32] to provide more information on the release mechanism. 

## 3. Results

### 3.1. Influence of the Initial Synthesis Conditions on the Drug-Free PLGA50 Particles

The size, size distribution and the structure of the PLGA50-based particles can be tuned with the optimization of the preparation parameters. To highly control the hydrodynamic diameter of the formed particles the fabrication of the PLGA drug carrier particles was carried out by using different organic solvents (acetone or DMSO), two types of stabilizers (PLUR or TWEEN) and various polymer concentrations. The measured diameters with the polydispersity indexes determined by DLS are shown in Figure 1 and Appendix A. 

The results of the concentration-dependence of the two stabilizers, where the amount of the PLGA was constant (c_PLGA_ = 5 mg/mL), demonstrate that the particle sizes are not changed significantly with their increasing concentrations when the acetone is used as organic phase (Figure 1A,B). In contrast, slightly increasing tendency in the diameters can be seen for use of DMSO. Moreover, we can clearly state that the application of DMSO results in significantly lower values (d~60–80 nm) than it was observed for acetone (d~110–120 nm). These experimental results highly indicate that the quality of the organic phase has an important effect on the formation of PLGA50 carrier particles. 

Based on the results presented on Figure 1A,B, 0.1 mg/mL TWEEN and PLUR stabilizer concentration was applied for further studies. After optimization of the stabilizer concentration, the effect of the concentration of the PLGA50 macromolecules was also studied on the particle formation (Figure 1C,D). It can be concluded that dominantly higher hydrodynamic diameter values can be measured by increasing of the PLGA amount from 2.5 mg/mL to 15 mg/mL, independently from the type of the stabilizers and the organic solvents. The values of the particle size are varied in the range of ~100–160 nm for PLUR and between ~100–150 nm in case of TWEEN in acetone. The ~10 nm difference between the two stabilizers is presumably due to their different molecular weights (M_w,TWEEN_ = 1228 Da, M_w,PLUR_ = ~12,600 Da). In case of the DMSO-containing systems, similar tendency is observed, but in a smaller particle size range (~60–95 nm for PLUR and ~50–90 nm for TWEEN). The polydispersity indexes (Appendix A) are not changed measurable. 

To characterize their colloid stability, the effect of NaCl concentration (c_NaCl_ = 0–0.16 M) was also studied on the change of the particle size in the absence and in the presence of the stabilizers. The Figure 2 shows that the increasing of the NaCl concentration results larger particle size and polydispersity, while the ζ-potential values are decreased. 

The aggregation of the particles can be observed at higher amount of NaCl. This observation clearly proves that the PLGA50 carrier particles, without the use of stabilizers, have an electrostatic stability, and due to this, it has a significant sensitivity to the change of the ionic strength. We also confirmed that, higher aggregation tendency can be observed in case of the application of DMSO. After the characterization of the stabilizer-free PLGA50 NPs, we also determined the NaCl effect in presence of TWEEN (Figure 3) and PLUR (Appendix A) stabilizers (c_stabilizer_ = 0.1 mg/mL).

In the TWEEN/PLGA50 systems, the hydrodynamic diameter is slightly increased with the NaCl concentration to a constant value (in acetone: d_DLS_~220 nm; in DMSO: d_DLS_~255 nm), where the dominant aggregation of the carriers cannot be detected (Figure 3). Due to the added NaCl, the ζ-potential is changed from −27 mV (c_NaCl_ = 0 M (in acetone) and −23 mV (c_NaCl_ = 0 M in DMSO) to ~0 mV, independently from the organic phase. Most probably the negatively charged carboxylate groups of PLGA50 rapidly compensate with Na^+^-ions, thus the electrostatic stability disappears in the presence of the inert salt, which causes the increasing of the particle size. However, the TWEEN provides steric stability to the systems, thus preventing the further aggregation. The adsorption of the surfactant is more favored to a near-neutral surface than on a surface that is denser in charge. Similar changes can be seen at the PLUR: d_DLS_ = 116 nm, ζ = −31 mV (c_NaCl_ = 0 M) and d_DLS_ = 219 nm, ζ = 2.5 mV (c_NaCl_ = 0.16 M) with acetone organic phase; d_DLS_ = 70 nm, ζ = −29 mV (c_NaCl_ = 0 M) and d_DLS_ = 251 nm, ζ = −1.2 mV with DMSO organic phase (Appendix A).

### 3.2. Characterization of the KP-Loaded PLGA NPs

Considering the previous results presented in Section 3.1, the following PLGA50 and stabilizer concentrations were used for encapsulation of KP as a model active compound: c_PLGA50_ = 10 mg/mL; V_PLGA50_ = 0.5 mL; c_PLUR/TWEEN_ = 0.1 mg/mL; V_PLUR/TWEEN_ = 5 mL. The increasing of the particle size was lower in the presence of the NaCl, when acetone was used, thus we applied this organic solvent for preparation of PLGA50/KP NPs. Particles were fabricated at different KP concentrations (c_KP_ = 0–15 mg/mL) and the size, size distribution, encapsulation efficiency (EE%) and the drug loading (DL%) were determined in every cases. The results can be seen in Table 1. The yield of the solid carrier NPs was 60–70% in every case, compared to the initial component mass. Particle size of the PLUR- and TWEEN-stabilized PLGA50/KP NPs is increased with KP concentration confirming the successful encapsulation of the drug. The EE% and the DL% are also increased in presence of PLUR, while maximum DL% is reached at c_KP_ = 10 mg/mL (EE%~61.9; DL%~38.2) when TWEEN is used as stabilizer. In addition, we found that the amount of the encapsulated KP cannot be changed indefinitely with the drug concentration, because the particles aggregate after few days above a given concentration (c_KP_ = 5 mg/mL) (Appendix A). To determine the stability of the drug-free and KP-loaded carriers in detail, the hydrodynamic diameter of the samples was monitored for 25 days by DLS. The data are summarized in Appendix A. The particle size is changed slightly to the lower values, probably, due to the degradation as well as the hydrolysis of the carriers and the decreasing of the KP content. To follow the kinetic of the decreasing of the hydrodynamic diameters, the data were fitted by zero order formulation. The determined rate constants are k = 0.884 nm/h (0 mg/mL KP) and k = 1.523 nm/day (5 mg/mL KP) for PLUR and k = 0.862 nm/day (0 mg/mL KP) and k = 1.017 nm/day (5 mg/mL KP) for TWEEN. Using these, the KP was also encapsulated in different PLGA (PLGA50; PLGA65 and PLGA75) at c_KP_ = 5 mg/mL, where similar data was observed, independently to the lactide:glycolide ratio and the applied stabilizer (Appendix A).

The morphology of the drug-free and a KP-loaded PLGA50 NPs with TWEEN and PLUR was characterized by TEM. The registered images in Figure 4. show that spherical structure can be identified in the absence of KP. However, it can be concluded, formation of the core-shell structure is highly confirmed in the presence of KP drug, where the stabilizers are formed shell on the KP-contained PLGA core (Figure 4C,D). Thinner layer can be observed in case of the TWEEN 20, presumably due to the lower molecular weight. This proves that the core-shell structure can be appeared with hydrophobic drugs. The average diameters measured by TEM image are in good agreement with DLS results.

The structure of the drug-loaded particles was studied by FT-IR spectroscopy and DSC measurements. Shift in the IR-spectra of the KP-loaded PLGA50 cannot be observed compared to the spectrum of the pure PLGA50 samples, which suggests that the formation of strong interaction is not realized, therefore its role is negligible during the dissolution of the active compound (Figure 5A,B). 

The appearance of the stretching vibration of the C=O (ν = 1695 cm^−1^) and the keto (ν = 1655 cm^−1^) on the spectra (gray dotted line), moreover the 715 cm^−1^, 703 cm^−1^ and 690 cm^−1^ bands of deformation vibration of the KP groups in the fingerprint region (ν ≤ 1500 cm^−1^) confirm the presence of encapsulated drug [33,34,35]. Similar changes can be detected in case of the PLGA65 and 75 (Appendix A). Based on the DSC curves, the endotherm melting point of the crystallic KP is 88 °C, and the degradation can be seen in 300–400 °C temperature range with 377 °C degradation peak maximum (T_d,max_) (Figure 5C,D) [36]. The endothermic T_d,max_ of the PLGA50 is decreased from 357 °C to 349 °C (PLUR/PLGA50/KP) and 347 °C (TWEEN/PLGA50/KP) with the decreasing of the KP concentration, from which the presence of a weak hydrophobic interaction can be assumed. 

### 3.3. In Vitro Drug Release Study

In case of well-defined core-shell structure, the retention can be tuned with thickness and compactness of the shell, in contrast to the homogenous distribution of the carrier components, where only the polymer matrix influences the dissolution of the encapsulated drugs. Taking this into account, the dissolution processes were followed to investigate the release properties of the core-shell structured TWEEN/PLGA/KP and PLUR/PLGA/KP carriers. During the studies, the measurement conditions as the pH of the phosphate buffer (pH = 6–8), the temperature (T = 25–45 °C) and the monomer ratios of the PLGA (50:50, 65:35 and 75:25 lactide:glycolide ratio) were systematically changed (Figure 6, Figure 7 and Appendix A). The reference was the solid KP in every cases. The temperature-dependence examinations point out that increase in the temperature results in higher drug release which is in line with increasing solubility. Moreover, we can establish that the PLUR/PLGA50/KP particles (Figure 6A) have lower retention than the TWEEN/PLGA50/KP systems (Figure 6B), which is based on the fact that the TWEEN molecules having relatively lower molecular weight form a more compact adsorption layer around the PLGA particles than the PLUR with larger M_w_ thus the drug dissolution is more regulated. 

To analyze the contribution of the carriers to the retention of the drug, the values of the c_t_/c_∞_ of the PLUR/PLGA50/KP NPs and TWEEN/PLGA50/KP NPs at 420 min were subtracted from the solid KP values. The Figure 6D. shows, while large effect cannot be seen in TWEEN/PLGA50/KP NPs, until then the retention of the PLUR/PLGA50/KP NPs is reduced from 12.7% (25 °C) to 6.0% (45 °C). Based on these, it can be concluded, the PLUR-stabilized carriers have temperature-sensitive release properties, presumably thanks to the surfactant. Significant change in the release profiles cannot be observed at the pH-dependence measurements, where the retention of the PLUR/PLGA50/KP NPs and the TWEEN/PLGA50/KP NPs are ~20% and ~70%, respectively (Appendix A). It is confirmed, the pH shift is not influenced the dissolution of the KP.

To confirm the dominant effect of the monomer ratio in the copolymer, the KP release from PLGA50-, PLGA65- and PLGA75-based colloidal carriers was also studied at pH = 7.4 in PBS medium at 37 °C to model physiological conditions. According to the results, which are shown in Figure 7, the amount of the liberated KP is increased for both the PLUR (c_t_/c_∞_ (%) = 77.8 (PLGA50), c_t_/c_∞_ (%) = 83.0 (PLGA65) and c_t_/c_∞_ (%) = 89.5 (PLGA75)) and the TWEEN (c_t_/c_∞_ (%) = 19.6 (PLGA50), c_t_/c_∞_ (%) = 33.3 (PLGA65) and c_t_/c_∞_ (%) = 62.2 (PLGA75)) stabilizers. 

Similar to our previous research work on the (±)-α tocopherol (TP)-loaded PLGA NPs, the higher lactide content promotes the release of the drug into the buffer medium [30]. However, the encapsulation efficiency of the KP is not changed systematically with the monomer ratios in contrast to the TP-contained systems, where this effect can be influenced the dissolution processes. Therefore, these results are clearly indicated that exclusively the lactide to glycolide ratios have role in the drug retention. The release rate is higher at the TWEEN/PLGA/KP NPs, compared to the PLUR-stabilized systems, thus the influence of the lactide content can be regulated by the optimalization of the surfactants.

For the determination of the kinetic parameters, the release curves of the KP were fitted by different kinetic equations (First order, second order, Weibull, Higuchi and Korsmeyer-Peppas formulations) with nonlinear regression The Weibull and the Second order models gave the best fit at the PLUR/PLGA/KP NPs, while the Weibull and the Korsmeyer-Peppas kinetic formulations showed the best results for TWEEN/PLGA/KP NPs. The appropriate kinetic data are shown in Appendix A. The diffusion-dissolution index (n) from Korsmeyer-Peppas model can be determined the erosion and diffusion contribution to dissolution processes (n = 0.42 (diffusion-controlled mechanism), n = 1 (Case II relaxation-controlled mechanism) and 0.42 ≤ n ≤ 1 (both of them) in case of the sphere shape). At the PLUR/PLGA/KP NPs, the diffusion dissolution index is n = 0.2503 (PLGA50), n = 0.2811 (PLGA65) and n = 0.2397 (PLGA75), which low values can be explained with the polydispersity of the particles [37]. With this in mind, it is confirmed that the KP release from the carriers has diffusion-controlled feature. The TWEEN/PLGA/KP NPs have higher n values, thus the appearance of the relaxation-controlled mechanism is proved. This can be presumably related to the erosion of the TWEEN layer, which is confirmed by the difference between the release profiles of the TWEEN- and PLUR-stabilized PLGA carriers. 

## 4. Conclusions

In our work, we clearly highlighted the effect of the main preparation conditions (like component ratios, NaCl concentration or type of the organic phase and the stabilizer structure) on the size, structure, the encapsulation efficiency, and the controlled drug release of KP-loaded PLGA particles. Considering the DLS measurements, we can be stated that the preparation of the PLGA carriers is a designable process, where the hydrodynamic diameter of the particles can be systematically changed by the concentration of the polymers and the stabilizers. It can be successfully concluded that ~50 nm smaller particle size can be achieved with using DMSO instead of acetone using the nanoprecipitation protocol. The stability experiments highlighted that, the stabilizer-free PLGA NPs have electrostatic stability, and it disappears at 0.02 M NaCl concentrations. However, due to the steric stability, this effect can be controlled with the use of non-ionic PLUR and TWEEN surfactants having non-toxic features. During the encapsulation of the KP, it can be confirmed, that the formation of the core-shell structured KP-loaded PLGA NPs is possible by the nanoprecipitation method. The optimal initial drug concentration was determined at 5 mg/mL with 7–8% drug loading, because the aggregation of the systems can be observed at higher concentration. The time-dependence measurements of particle size for the drug-free and the 5 mg/mL KP-loaded PLGA50 NPs pointed out that the diameters are continuously decreased, presumably, due to the degradation of the carriers. The determined in vitro release profiles indicate that ~60% higher retention can be achieved in case of the TWEEN, regardless of the pH and the temperature of the medium. We highlighted that the type of the stabilizer has more influence on the dissolution processes. Namely, the TWEEN molecules having relatively lower molecular weight form a more compact adsorption layer around the PLGA particles than the PLUR with larger M_w_ thus the drug dissolution is more regulated. 

The release curves were fitted by different kinetic models, where the value of the diffusion -dissolution index (n) from the Korsmeyer-Peppas formulation is pointed out that the erosion—controlled features in addition to the diffusion-controlled mechanism are also present in the TWEEN—stabilized systems, while the release of the KP from the PLUR/PLGA/KP NPs only has diffusion properties. 

## Figures and Tables

**Figure 1 pharmaceutics-15-01355-f001:**
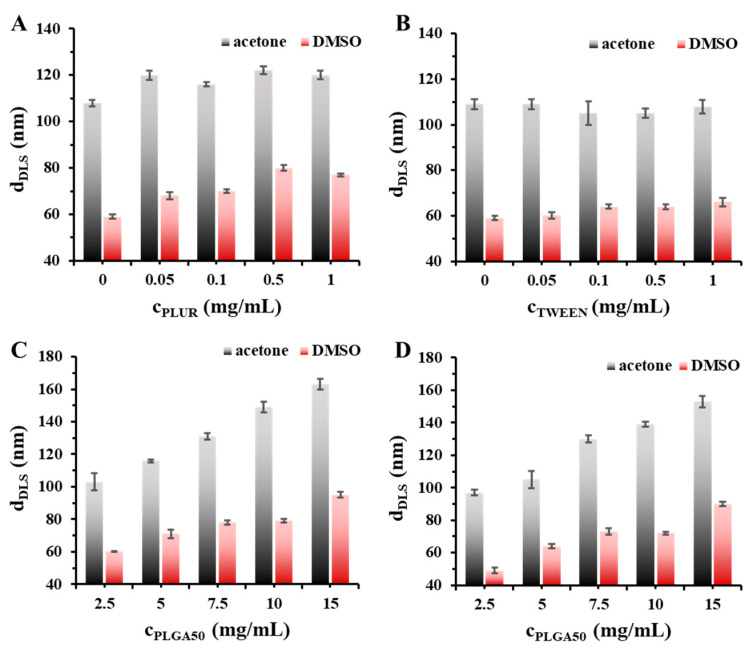
Hydrodynamic diameter of the drug-free PLGA50 particles at different stabilizer concentrations (**A**,**B**) (c_PLGA50_ = 5 mg/mL; V_stabilizer_ = 5 mL; 1:10 organic solvent: water ratio) and at various PLGA50 concentrations (**C**,**D**) (c_stabilizer_ = 0.1 mg/mL; V_stabilizer_ = 5 mL; 1:10 organic solvent: water ratio) (**A**,**C**: PLUR stabilizer; **B**,**D**: TWEEN stabilizer; acetone and DMSO are marked with black (always the left column at each concentration) and red columns, respectively, in the online form).

**Figure 2 pharmaceutics-15-01355-f002:**
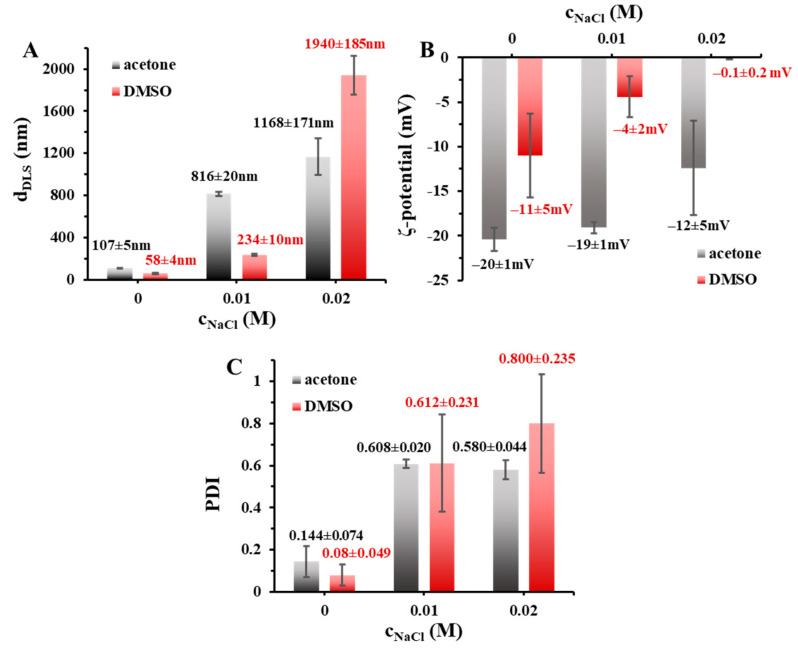
(**A**) Hydrodynamic diameter, (**B**) ζ-potential and (**C**) polydispersity (PDI) of the drug- and stabilizer-free PLGA50 colloidal particles at different NaCl concentrations (c_PLGA50_ = 5 mg/mL; V_water_ = 5 mL; 1:10 organic solvent: water ratio; acetone and DMSO are marked with black (always the left column at each concentration) and red column, respectively, in the online form).

**Figure 3 pharmaceutics-15-01355-f003:**
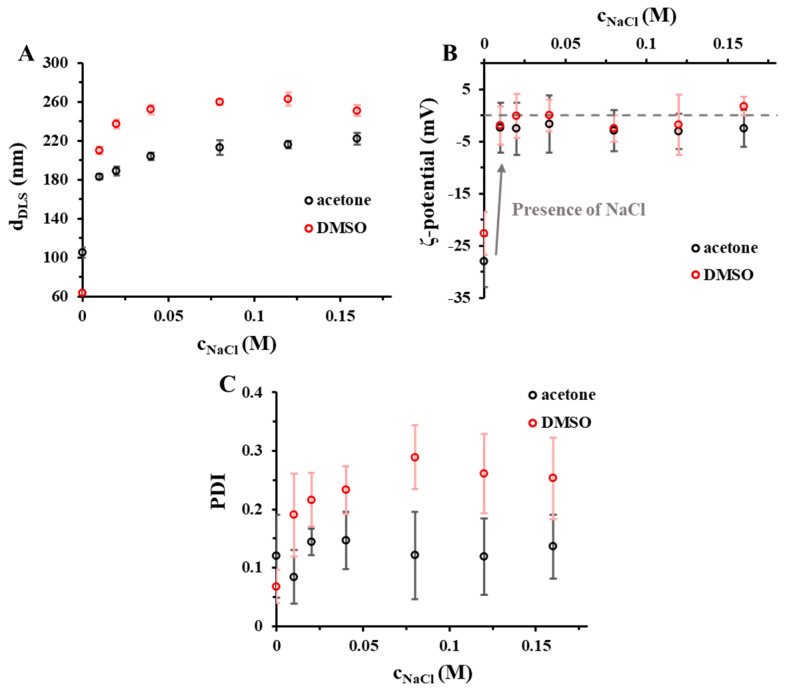
(**A**) Hydrodynamic diameter, (**B**) ζ-potential and (**C**) polydispersity (PDI) of the drug-free TWEEN/PLGA50 colloidal particles at different NaCl concentrations (c_PLGA50_ = 5 mg/mL; c_stabilizer_ = 0.1 mg/mL; V_stabilizer_ = 5 mL; 1:10 organic solvent: water ratio) (acetone and DMSO are marked with black and red ◯ symbol, respectively, in the online form).

**Figure 4 pharmaceutics-15-01355-f004:**
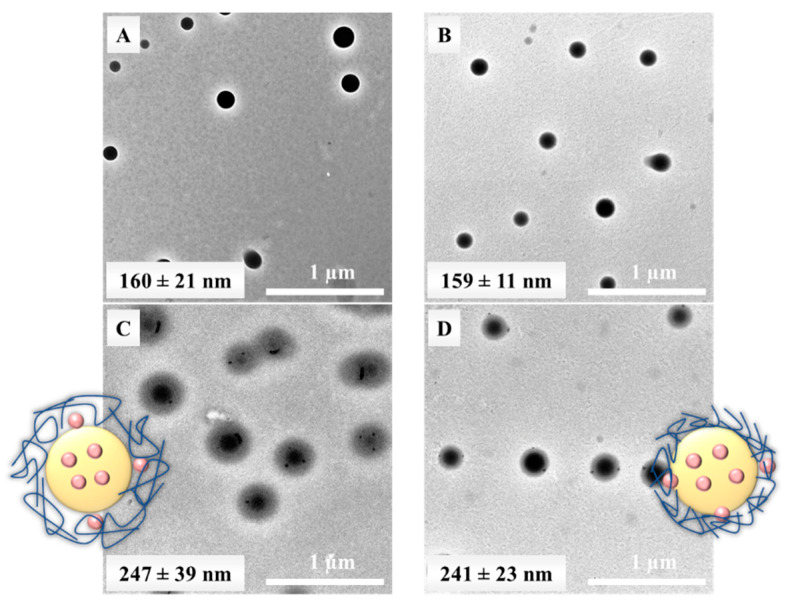
Representative TEM images of the (**A**,**B**) drug-free and the (**C**,**D**) KP-loaded PLGA50 NPs (c_PLGA50_ = 10 mg/mL; c_KP_ = 0 or 5 mg/mL, c_PLUR/TWEEN_ = 0.1 mg/mL; (**A**,**C**) PLUR stabilizer; (**B**,**D**) TWEEN stabilizer). Schematic not scale representation of the KP-loaded PLGA NPs core-shell structure (inset).

**Figure 5 pharmaceutics-15-01355-f005:**
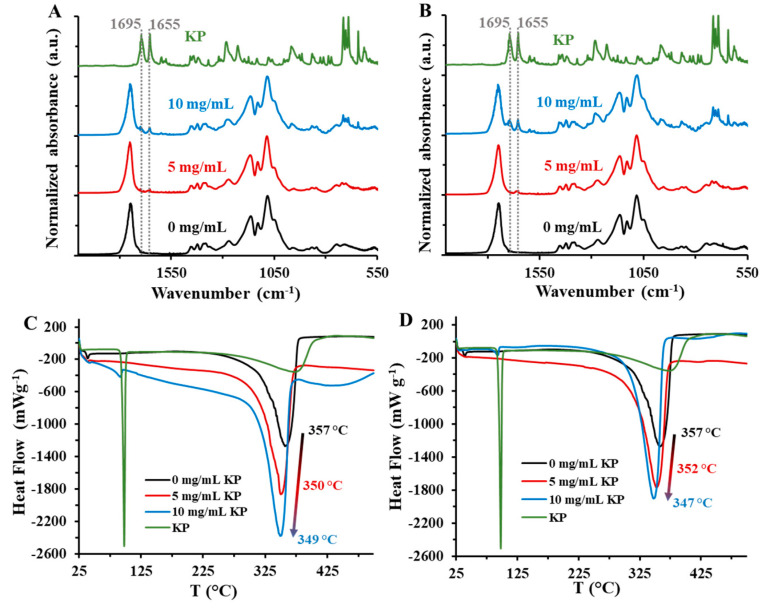
(**A**,**B**) FT-IR spectra and the (**C**,**D**) DSC curves of the the solid KP and the KP-loaded PLGA50 NPs at different drug concentrations (c_PLGA50_ = 10 mg/mL; c_KP_ = 0–10 mg/mL, c_PLUR/TWEEN_ = 0.1 mg/mL; (**A**,**C**) PLUR stabilizer; (**B**,**D**) TWEEN stabilizer).

**Figure 6 pharmaceutics-15-01355-f006:**
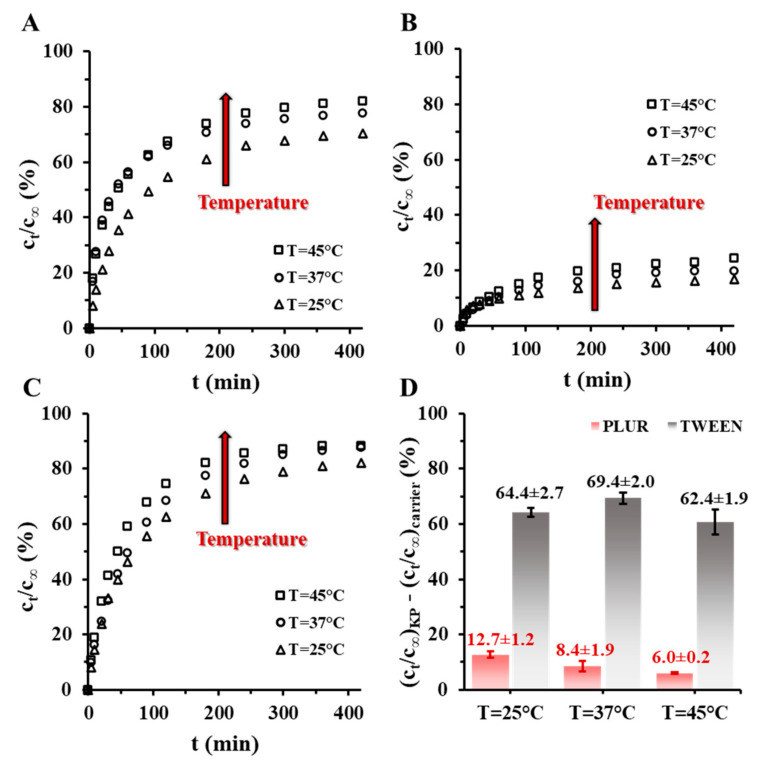
The dissolution curves of the (**A**) PLUR/PLGA50/KP NPs, (**B**) TWEEN/PLGA50/KP NPs and the (**C**) solid KP at different temperature (T = 25–45 °C; pH = 7.4 PBS medium; 0.9% NaCl), and (**D**) the retention of the PLGA50/KP NPs corrected with released drug value of the solid KP at 420 min.

**Figure 7 pharmaceutics-15-01355-f007:**
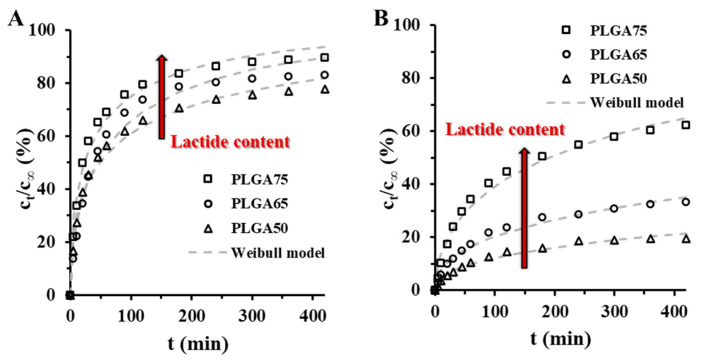
Release profiles and the fitted Weibull kinetic model (dotted gray line) of the (**A**) PLUR/PLGA/KP NPs and (**B**) the TWEEN/PLGA/KP NPs in PBS medium at different lactide to glycolide ratios (T = 37 °C; pH = 7.4; 0.9% NaCl, t = 0–420 min).

**Table 1 pharmaceutics-15-01355-t001:** The hydrodynamic diameter (d_DLS_), polydispersity (PDI), encapsulation efficiency (EE%) and the drug loading (DL%) of the KP-loaded PLGA50 NPs (c_PLGA50_ = 10 mg/mL; c_PLUR/TWEEN_ = 0.1 mg/mL, V_PLUR/TWEEN_ = 5 mL; 1:10 organic solvent: water ratio).

	c_KP_ (mg/mL)	d_DLS_ ± SD (nm)	PDI ± SD	EE ± SD (%)	DL ± SD (%)
**PLUR**	0.0	149 ± 3	0.144 ± 0.054	-	-
	2.5	195 ± 2	0.087 ± 0.049	14.5 ± 1.1	3.5 ± 0.3
	5.0	212 ± 6	0.106 ± 0.074	18.2 ± 1.4	8.3 ± 0.6
	10.0	222 ± 4	0.101 ± 0.015	37.0 ± 3.7	26.9 ± 2.0
	15.0	226 ± 5	0.118 ± 0.055	43.9 ± 5.3	39.6 ± 2.9
**TWEEN**	0.0	139 ± 2	0.076 ± 0.030	-	-
	2.5	187 ± 3	0.126 ± 0.051	12.2 ± 0.9	3.0 ± 0.2
	5.0	198 ± 4	0.131 ± 0.057	14.3 ± 1.1	6.7 ± 0.5
	10.0	208 ± 2	0.140 ± 0.055	61.9 ± 7.3	38.2 ± 2.8
	15.0	225 ± 9	0.203 ± 0.076	28.3 ± 2.9	29.7 ± 2.2

## Data Availability

Data is contained within the article and Appendix A.

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
