# Peer review of "Core-Shell Structured PLGA Particles Having Highly Controllable Ketoprofen Drug Release"

_pharmaceutics, 2023, doi:10.3390/pharmaceutics15051355_

Round 1

Reviewer 1 Report

In this manuscript, the authors reported a core-shell structured PLGA particles having highly controllable ketoprofen drug release. The work was very systematic and the results are quite impressive. I think it is publishable in pharmaceutics. A few minor questions.

1. You highlighted the" Core-shell structured "in the title, but did not provide a detailed explanation and representation in the article.

2. Because no animal experiments have been conducted in this article, the biological safety of the preparation needs to be verified. The biosafety of PLURONIC and TWEEN20 should be confirmed by citing the literature in the introduction section.

3. Breaking the ester bond will lead to the degradation of PLGA, and the degree of degradation varies with the monomer ratio. The greater the proportion of glycolide, the easier it is to degrade. I think PLGA65 should degrade faster than PLGA75 in Figure 7.

Reviewer 2 Report

In the present manuscript entitled “Core-shell structured PLGA particles having highly controllable ketoprofen drug release”, the authors have prepared PLGA particles, choosing ketoprofen as a model molecule. Moreover, to control drug release, two different stabilizers were used.

Though the studies are well designed and performed, there are some queried that need to be addressed before being considered for publication.

To me the study appears incomplete, and it would be too early to publish. In order to make the study complete the authors should include the in vitro studies along with in vivo efficacy.

In what ratio w/w the amount of surfactant was added with respect to the amount of polymer? Is it possible that it is a high amount, perhaps toxic? For this reason, in vitro studies are needed to evaluate that the systems are cytocompatible and do not cause cell damage. Furthermore, the anti-inflammatory activity of the encapsulated drug compared to the free drug must be tested, to show the efficiency of this system. At the same concentrations, empty system should be tested to confirm that it is non-toxic.

For a complete technological characterization, In Table 1 authors are suggested to add the yield.

The nanoparticle purification procedure used by the authors is questionable. In lines 108-109 the authors write that the nanoparticles were separated by a single centrifugation at 12000 rpm for 15-20 min. Being that the NPs have a very small diameter (as shown in table 1, line 248), was a single centrifuge enough? Is the supernatant clear? In the literature there are numerous manuscripts in which more centrifuges are carried out for the purification of the nanoparticles, at higher rpm and at low temperatures. The authors need to state the rationale of present method.

Line 146: please add unit of measurement of the membrane cut-off.

The morphology should be studied before and after freeze-drying.

Reviewer 3 Report

The manuscript by Varga et al. describes the synthesis and the modulation of PLGA core-shell particles by varying different parameters. While the manuscript might be of interest for readers, there are some issues that must be addressed.

The introduction should be modified. At this point, it is more likely a Discussion rather than an Introduction.

There is no Discussion section, which is very important for presenting the results more clearly.

Regarding the design of the study, I have some concerns regarding the use and the removal of Tween and Pluronic for biomedical applications. Since they are surfactants, they affect the cell membrane; thus, they could be seriously toxic for the envisaged applications. How did the authors consider this issue?

English should be thoroughly checked. The manuscript is difficult to read at this point.

Round 2

Reviewer 1 Report

I'm glad my suggestion can help you, but this manuscript does not provide any direction for future research or the next steps after the completion of this study. Please consider providing some guidance in this regard. 

Reviewer 3 Report

The authors did not feel the need to address my concerns.

I still do not think that the manuscript is publishable at this point. The study involves the use of surfactants which, given their molecular structure, have a tendency to disrupt the cellular membrane. Considering the scope of the Pharmaceutics journal, the study is not suited for publishing at this point (there are no cell assays that could prove the suitability of their polymeric systems to be applied in the biomedical field). Additionally, the abstract states "the design of a biocompatible colloidal carrier particles" - how do you know that your colloidal carrier particles are biocompatible? Citing some references is definitely not enough. If the authors do not want to address these concerns, I suggest finding a more suitable journal that solely focuses on the material aspect and not on the interaction with cells.

Additionally, the presentation of the manuscript is not appropriate, it is very difficult to follow starting from the Introduction, which should easily introduce the reader into the subject.

Round 3

Reviewer 3 Report

The authors have stated that they cannot perform in vitro testing to determine the suitability of their drug delivery systems for biomedical applications.